# Comprehensive analysis of the lysine succinylome in fish oil-treated prostate cancer cells

Yifan Jiang[1],*, Chao He[1],* (ORCID), Haokai Ye[2], Qianhan Xu[2], Xiuyuan Chen[2], Ying Chen[1], Jianying Sun[2], Sonia Rocha[3] (ORCID), Mu Wang[1,2] (ORCID)

Prostate cancer (PCa) poses a significant health threat to males, and research has shown that fish oil (FO) can impede PCa progression by activating multiple mitochondria-related pathways. Our research is focused on investigating the impact of FO on succinylation, a posttranslational modification that is closely associated with mitochondria in PCa cells. This study employed a mass spectrometry-based approach to investigate succinylation in PCa cells. Bioinformatics analysis of these succinylated proteins identified glutamic–oxaloacetic transaminase 2 (GOT2) protein as a key player in PCa cell proliferation. Immunoprecipitation and RNA interference technologies validated the functional data. Further analyses revealed the significance of GOT2 protein in regulating nucleotide synthesis by providing aspartate, which is critical for the survival and proliferation of PCa cells. Our findings suggest that FO-dependent GOT2 succinylation status has the potential to inhibit building block generation. This study lays a solid foundation for future research into the role of succinylation in various biological processes. This study highlights the potential use of FO as a nutrition supplement for managing and slowing down PCa progression.

## Introduction

Worldwide, prostate cancer (PCa) is a prevalent cancer type and is the second leading cause of cancer-related deaths in men, with 99,000 new cases increasing in the US from 2014 to 2019 (Orang et al, 2019; Mishra, 2021; Siegel et al, 2023). Numerous studies have shown that early intervention in nutrition and dietary patterns may offer promising approaches to reduce the risks of developing PCa or to slow down its progression (Lin et al, 2015). The consumption of certain oils, such as saturated fats and plant-based $\omega$-9 fatty acids primarily composed of oleic acid (OA), has been associated with an increased incidence of PCa (Elkacmi et al, 2016; Allott et al, 2017;

Aldaw et al, 2018; Liotti et al, 2018; Wang et al, 2022). On the other hand, fish oil (FO), which is primarily composed of omega-3 polyunsaturated fatty acids ($\omega$-3 PUFAs), has been shown to reduce the risk of developing PCa (Williams et al, 2011; Bassett et al, 2013; Richman et al, 2013; Pan et al, 2017). The major components of $\omega$-3 PUFAs in FO are eicosapentaenoic acid (EPA) and docosahexaenoic acid (DHA) (Hull, 2011). Currently, the molecular mechanism underlying the metabolism of these unsaturated fatty acids in PCa remains elusive and requires further elucidation.

Our previous proteomic and phosphoproteomic profiling studies indicated that $\omega$-3 PUFAs trigger multiple pathways in PCa cells, including survival stress induction, attenuation of pro-inflammatory molecules, and suppression of specific growth signals (Zhao et al, 2016a, 2016b). One of the most significant findings indicates that FO can disturb mitochondria function in PCa cells, which shows potential therapeutic values (Ashton et al, 2018; Orang et al, 2019).

From the classic Warburg effect theory to the modern understanding of cancer metabolism, abundant evidence has demonstrated that mitochondria play a pivotal role in supporting cancer progression (Warburg et al, 1927; Vaupel et al, 2019; Benny et al, 2020). Therefore, it is crucial to investigate the changes in mitochondrial physiological and chemical properties after FO treatment (Ksiezakowska-Lakoma et al, 2014; Benny et al, 2020; Luo et al, 2020). In fact, in 2015, research has shown that the primary function of the electron transport chain (ETC) in cancer cells is to provide aspartate rather than ATP (Birsoy et al, 2015). Cells typically take up only a tiny fraction of aspartate from the cell culture media, and the majority is synthesized via the malate–aspartate shuttle (Cheng et al, 2018; Sullivan et al, 2018). Glutamic–oxaloacetic transaminase 2 (GOT2) is one of the key components operating in the malate–aspartate shuttle, which enhances metabolic fitness by converting malate into aspartate (Wu et al, 2021). Many studies have revealed that aspartic acid is a fundamental precursor to nucleotides (Safer, 1971; Garcia-Bermudez et al, 2018). Therefore, alterations to GOT2 can modify its properties, potentially impacting the generation of aspartate and nucleotides.

[1]Academy of Pharmacy, Xi'an Jiaotong-Liverpool University, Suzhou, China   [2]Department of Biological Sciences, Xi'an Jiaotong-Liverpool University, Suzhou, China [3]Department of Molecular Physiology and Cell Signaling, Institute of Systems, Molecular and Integrative Biology, University of Liverpool, Liverpool, UK

Correspondence: mu.wang@xjtlu.edu.cn
*Yifan Jiang and Chao He contributed equally to this work

In this study, a new type of posttranslational modification (PTM) closely related to mitochondria function has become the research focus. In 2011, succinylation was identified using a high throughput mass spectrometry method and confirmed by Western blot analyses and in vivo labeling (Zhang et al, 2011). Interestingly, succinylation is lysine (K)-specific, which allows for shared modification sites with other PTMs such as acetylation (Weinert et al, 2013; Ye & Li, 2022). This suggests that cells can modulate protein function by switching between acetylation and succinylation at specific lysine residues (Ali et al, 2019). However, succinylation causes more dramatic protein structure changes than acetylation due to its bulkier size, as the addition of 100.186 Da is more significant than that of 42 Da in acetylation. Furthermore, succinylation brings a more negative charge that flips the unmodified lysine residue from positive to negative (Zhang et al, 2011). The alteration of substrate size and charge induced by succinylation has been demonstrated to impact protein–protein interactions, thereby promoting tumor growth (Li et al, 2020; Mu et al, 2021). Although protein levels remain unchanged, differences in succinylation levels may significantly impact protein function and pathway in PCa cells.

This study conducted a global succinylomic profile in the PC-3 cell line after FO, OA, and EtOH treatments. The following bioinformatics analysis enriched the succinylated proteins detected by mass spectrometry based on their function or location, which highlights the succinylated GOT2 protein because of its role in promoting the generation of metabolic precursors in PC-3 cells. To confirm the ability of EPA to induce GOT2 succinylation, immunoprecipitation (IP) was performed on PCa cell lysate to enrich for GOT2 protein and assess its succinylation status. Subsequent aspartate quantification assays validated the Gene Ontology (GO) and Kyoto Encyclopedia of Genes and Genomes (KEGG) findings by demonstrating that normal GOT2 is responsible for generating increased levels of aspartate required for cellular proliferation. It is postulated that the $\omega$-3 PUFAs-dependent succinylation in GOT2 protein leads to a reduction in nucleotide production, which fails to meet the high proliferative demands of PCa cells. These findings thus provide valuable insights into $\omega$-3 PUFAs metabolism in PCa cells and can be useful for further functional studies of lysine succinylation.

# Results

### $\omega$-3 PUFA EPA inhibits PCa cell growth

To investigate whether EPA could modulate cell proliferation, clonogenic assay, growth curve, and EdU assay were conducted. Colony formation assays in both cell lines demonstrated a dose-dependent growth inhibitory effect of EPA (Fig 1A and B), with no colony formation observed at the highest EPA concentration (100 $\mu$M). In contrast, the groups treated with OA exhibited a more rapid proliferation compared with the control group. Besides, the cell proliferation pattern was visualized by growth curves generated from the counted cell numbers over a continuous 6–8 d period, which yielded results consistent with those obtained from colony formation assays (Fig 1C and D). The EdU assay further confirmed

that the proliferation rate of PCa cell lines was reduced to less than 75% after treatment with 100 $\mu$M EPA (Fig 1E and F), thus validating the inhibitory effect of EPA on PCa cell proliferation and survival. Moreover, FO treatment has been shown to have a selective impact on the survival of PCa cells (PC-3 and LNCaP), whereas not influencing the healthy prostate epithelium RWPE-1 cells (Fig S1).

### Global succinylomic analysis of human PCa cell lines treated with FO

The present study aimed to investigate the mechanisms underlying the inhibition of cell proliferation in PCa cells through the evaluation of succinylation level changes after 100 $\mu$M EPA treatment. The workflow of this study is illustrated in Fig 2A. A mass spectrometry-based global succinylation analysis was conducted to detect and quantify 946 succinylated peptides as shown in Fig 2B. Among these peptides, 125 (ranging from 6 to 22 amino acids) exhibited changes in succinylation level (Fig 2B and Table S1). Notably, 36% of these proteins had only one succinylation site, and HSPD1 was identified as the most modified protein, with nine Ksu sites (Fig S2).

Peptides with a $P$-value < 0.05 for any of FO-EtOH (FO-treated as the experimental group and EtOH-treated as the background control), OA-EtOH or FO-OA were considered to have significant changes in their succinylation level and were selected for further bioinformatics analysis. GO and KEGG were used for bioinformatics analysis (Fig 2D and E and Table S2). Table 1 presents partially the peptides from the FO-EtOH comparison with a $P$-value < 0.05 and a $|\log_2 FC| > 1.5$. In addition, the volcano plot generated from the FO-EtOH comparison is depicted in Fig 2C. Interestingly, both FO-OA and FO-EtOH groups revealed an increase in the overall succinylation level, resulting in the volcano plots inclining towards the right. Conversely, the OA-EtOH group exhibited a balance between up- and down-regulation of succinylation level (Figs S3 and S4).

### GO and KEGG enrichment analyses reveal the versatile function of GOT2 protein in PCa cell

The proteins containing the selected succinylated peptides were analyzed by bioinformatics approaches to investigate the pathways involved. GO analysis revealed that the most enriched pathway after FO treatment was associated with the generation of precursors for metabolites and energy (Fig 2D). According to the results of cellular component (CC) enrichment, succinylation occurs both intra- and extra-mitochondrially, with the mitochondrial compartment being identified as the primary site of succinylation (Fig 2D). This suggests a strong association between succinylation and mitochondrial function (Porporato et al, 2018; Vasan et al, 2020).

Furthermore, KEGG analysis identified several highly enriched pathways (Fig 2E). Carbon metabolism was found to be the most enriched pathway, followed by the biosynthesis of amino acids, the TCA cycle, glyoxylate and dicarboxylate metabolisms, and glycolysis/gluconeogenesis ($P$ < 0.01). The biosynthesis of amino acids (Fig 2E and F) involved 14 Ksu sites in 9 proteins, including GOT2, which is a vital member of the malate–aspartate shuttle (Kerk et al, 2022).

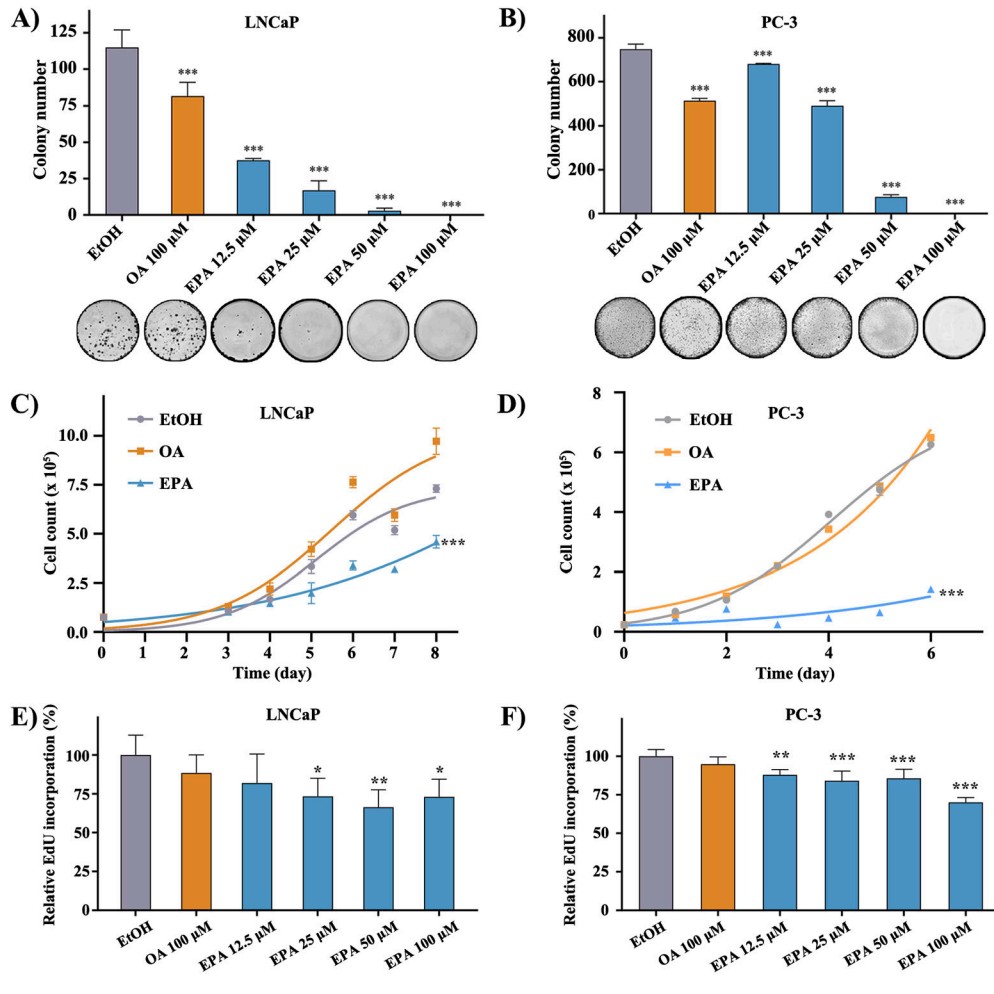

**Figure 1. FO treatment inhibits PCa cell proliferation.**
**(A, B)** Number of colonies in different treatment groups. Cell clusters containing more than 50 cells were considered colonies. The Count tool in Photoshop facilitates the counting process. **(C, D)** Proliferation curve of PCa cells treated with EtOH, OA or EPA. EPA is one of the active components of FO. The record period for the LNCaP cell line is 2 d longer because of its weak adherent ability compared with the PC-3 cell line. **(E, F)** EdU incorporation of PCa cells was measured 24 h after oil supplements. The solution contained EdU (5-ethynyl-2′-deoxyuridine) that structurally mimicked thymidine and was added to cells and incorporated into the DNA during the S phase.

## EPA promotes GOT2 succinylation

Roles of the GOT2 protein have been highlighted in Figs 2 and 3A shows the MS/MS spectrum of the peptide [154]DVFLPK[su]PTWGNHT-PIFR[170] of GOT2, confirming the succinylation in the K159 site of the GOT2 protein. To validate this MS result, immunoprecipitation (IP) was performed to pull down the GOT2 protein from a whole cell lysate and the succinylation change in the enriched GOT2 protein was detected by Western blot (WB) using a lysine succinylation motif antibody. IP results reveal a noticeable twofold increase of GOT2 succinylation level in the EPA-treated group compared with the OA and the control groups (Fig 3B and C).

## EPA reduces cellular aspartate content in PCa cells

To better understand the function of GOT2 in cancer cell growth and metabolism, we knocked down its level by RNAi technology. qRT-PCR and WB results all showed a successful expression-interference effect in both PCa cell lines (Figs 4A–D and S5). The aspartate quantification assay further indicates a decrease in aspartate level upon down-regulation of GOT2 (Fig 4E and F), and the growth curve reveals that the aspartate generated by GOT2 is used for cell proliferation. It is because

knockdown of GOT2 inhibits PCa cell growth, but interestingly, the growth rate can be brought back up to almost normal by feeding 20 mM aspartate in the media (Figs 4G and H and S6A and B).

Moreover, aspartate contents in PCa cells decreased after FO treatment, whereas it slightly increased in the OA-treated group (Fig 5A and B). Supplying FO-treated PCa cells with 20 mM aspartate can relieve the proliferation inhibitory effect (Fig 5C and D) but is not capable of rescuing the growth rate to the level of the EtOH group (Fig 1C and D). These results imply that the function of GOT2 protein may be affected by FO treatment through lysine succinylation, resulting in impaired aspartate production (Fig 5E).

## Discussion

PTMs play critical roles in modulating cellular processes (Farley & Link, 2009; Yin et al, 2019). Among more than 300 known PTMs, phosphorylation, acetylation, glycosylation, and methylation have been extensively studied (Zhao & Jensen, 2009; Ramazi et al, 2020; Jamal et al, 2021). Succinylation, which has a strong connection with mitochondria compared with other PTMs, has gained attention since its discovery in 2011. Despite its importance, succinylomic profiling studies

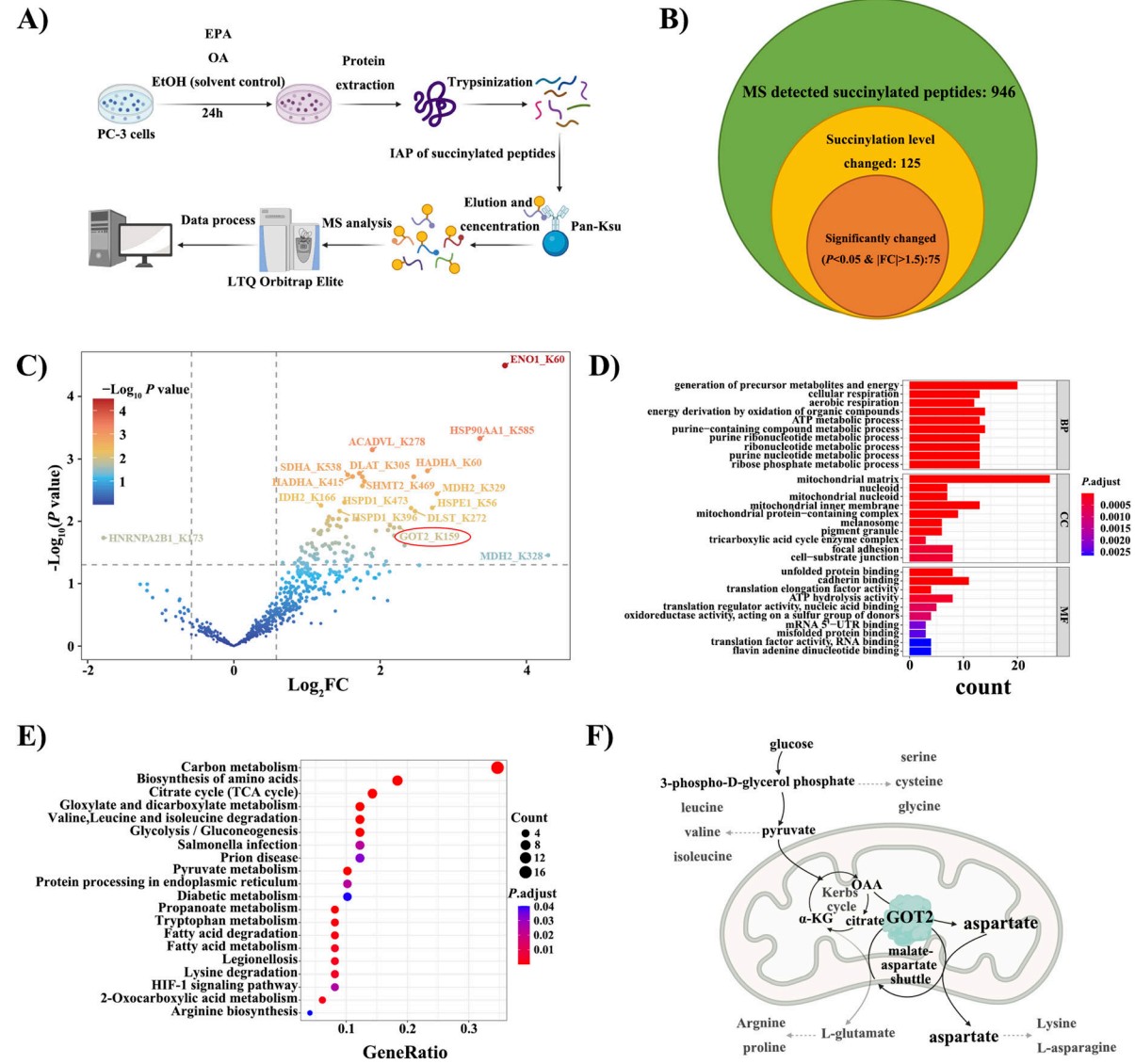

**Figure 2. Basic statistical summary and bioinformatics analysis of succinylated peptides and proteins.**
**(A)** Flowchart of the MS experiment. Succinylated peptides were enriched from digested peptides of the whole cell lysate. **(B)** The Venn diagram of MS results. In total, 946 succinylated peptides were detected, and 75 were significantly changed in the succinylation level. **(C)** A Volcano plot of detected succinylated peptides in the FO-EtOH group. The red ovary circle indicates one of the components in the malate–aspartate shuttle. **(D)** GO enrichment of succinylated proteins in three levels; BP (biological process), CC (cellular component), and MF (molecular function). Generation of precursor metabolites and energy, mitochondria matrix, and cadherin binding are most enriched with a small P-adjust value. **(E)** KEGG pathway enrichment of candidate peptides corresponding proteins by R studio platform. **(F)** Simplified *biosynthesis of the amino acid* pathway, drawn by BioRender (Created with BioRender.com, agreement number: PA25890XPP) adapted from the online KEGG color pathway search result. Two cycles in mitochondria represent the Krebs cycle and the malate–aspartate shuttle.

have mostly been done in prokaryotes that do not have mitochondria (Colak et al, 2013; Zhang et al, 2017; Dong et al, 2021). This study empowers us to investigate the differential protein succinylation levels in PCa cells under different physiological conditions.

In this study, a comprehensive succinylome was constructed after treatments with FO, OA, and EtOH in PC-3 cells, and the subsequent bioinformatics analysis yielded abundant information. The volcano plot of FO-EtOH and FO-OA are unsymmetrical and tilted to the right, demonstrating an overall up-regulation of succinylation level after FO treatment (Figs 2C and S3). The CC enrichment results of our study through GO analysis revealed that

succinylation mainly happens to mitochondrial proteins, which is likely because of the location of succinyl-CoA, a donor of succinyl groups in the Krebs cycle (Sreedhar et al, 2020). Subsequently, the GO-enriched pathways demonstrated a correlation between FO-induced GOT2 succinylation, subsiding in the malate–aspartate shuttle, and decreased precursor metabolites generation. The amino acid synthesis pathway enriched from the KEGG analysis further confirmed this correlation, indicating that GOT2 protein plays a crucial role in producing aspartic acid, which is required for PCa progression because it is the nitrogen donor for amino acid arginine (Reitzer, 2004; Walker & van der Donk, 2016).

**Table 1. Succinylated peptides with *P* value < 0.05 and |log$_2$FC| > 1.5 (|FC| > 2.8).**

| Uniport | Protein | Site | Sequence | log$_2$FC-FO-EtOH | *P*-value-FO-EtOH |
|---|---|---|---|---|---|
| P40926 | MDH2 | MDH2_K328 | EKMISDAIPELKASI**K**KGEDFVKTLK | 4.296960977 | 0.035099562 |
| P40926 | MDH2 | MDH2_K329 | KMISDAIPELKASIK**K**GEDFVKTLK | 2.775587977 | 0.003616751 |
| P61604 | HSPE1 | HSPE1_K56 | VLQATVVAVGSGSKG**K**GGEIQPVSVKVGDKV | 2.711728639 | 0.006076349 |
| P62826 | RAN | RAN_K37 | GKTTFVKRHLTGEFE**K**KYVATLGVEVHPLVF | 2.476389007 | 0.006889485 |
| P36957 | DLST | DLST_K272 | SNIQEMRARHKEAFL**K**KHNLKLGFMSAFVKA | 2.476389007 | 0.006889485 |
| P09622 | DLD | DLD_K267 | GIDMEISKNFQRILQ**K**QGFKFKLNTKVTGAT | 2.42263796 | 0.006142158 |
| P00505 | GOT2 | GOT2_K159 | FLQRFFKFSRDVFLP**K**PTWGNHTPIFRDAGM | 2.265663327 | 0.012598908 |
| P38646 | HSPA9 | HSPA9_K288 | LGGEDFDQALLRHIV**K**EFKRETGVDLTKDNM | 2.132607469 | 0.011641536 |
| B9A062 | MTHFD2 | MTHFD2_K44 | HLAAVRNEAVVISGR**K**LAQQIKQEVRQEVEE | 2.106933174 | 0.013880777 |
| P30405 | PPIF | PPIF_K190 | KEGMDVVKKIESFGS**K**SGRTSKKIVITDCGQ | 2.081270957 | 0.044075908 |
| P24752 | ACAT1 | ACAT1_K263 | VVVKEDEEYKRVDFS**K**VPKLKTVFQKENGTV | 1.905554416 | 0.027635774 |
| P25705 | ATP5F1A | ATP5F1A_K531 | VSQHQALLGTIRADG**K**ISEQSDAKLKEIVTN | 1.891490813 | 0.036596889 |
| P30405 | PPIF | PPIF_K183 | HVVFGHVKEGMDVVK**K**IESFGSKSGRTSKKI | 1.874526588 | 0.032672665 |
| P36957 | DLST | DLST_K277 | MRARHKEAFLKKHNL**K**LGFMSAFVKASAFAL | 1.807973185 | 0.033748449 |
| Q9NVI7 | ATAD3A | ATAD3A_K539 | PGQEERERLVRMYFD**K**YVLKPATEGKQRLKL | 1.757269115 | 0.002701385 |
| H0YDD4 | DLAT | DLAT_K305 | LLVRKELNKILEGRS**K**ISVNDFIIKASALAC | 1.715698805 | 0.001699704 |
| H3BNX3 | SQOR | SQOR_K173 | PEGFAHPKIGSNYSV**K**TVEKTWKALQDFKEG | 1.689186919 | 0.009285569 |
| P00558 | PGK1 | PGK1_K184 | AHRAHSSMVGVNLPQ**K**AGGFLMKKELNYFAK | 1.632933084 | 0.025817795 |
| H0YDD4 | DLAT | DLAT_K298 | DVNMGEVLLVRKELN**K**ILEGRSKISVNDFII | 1.617668691 | 0.036427758 |
| P49411 | TUFM | TUFM_K286 | RGTVVTGTLERGILK**K**GDECELLGHSKNIRT | 1.599133472 | 0.040784987 |
| P62937 | PPIA | PPIA_K28 | DGEPLGRVSFELFAD**K**VPKTAENFRALSTGE | 1.558619889 | 0.023483822 |
| P31040 | SDHA | SDHA_K538 | AAVFRVGSVLQEGCG**K**ISKLYGDLKHLKTFD | 1.555876117 | 0.001791026 |
| P11142 | HSPA8 | HSPA8_K128 | SFYPEEVSSMVLTKM**K**EIAEAYLGKTVTNAV | 1.553530112 | 0.043720757 |

Succinylated lysine (K) is in bold.

As the results from this study indicate, FO is capable of inhibiting PCa progression via multiple pathways and mechanisms. It is hypothesized that FO-dependent structure change and charge flip on the GOT2 K159 residue may induce unexpected repelling forces, thereby impacting aspartate production and resulting in the suppression of PCa cell proliferation. Given that mitochondrion plays a crucial role in nucleotide precursor production, a sufficient supply of aspartate is necessary for vigorous cell proliferation (Birsoy et al, 2015; Sullivan et al, 2015). In fact, this hypothesis is based on previous research that identified acetylation at the same site, indicating that K159 acetylation can enhance protein–protein interactions within the malate–aspartate shuttle and promote tumor growth (Yang et al, 2015).

Our study also showed that feeding GOT2 knockdown cells in the media with 20 mM aspartate can compensate for the aspartate deficiency caused by GOT2 knockdown. The chosen concentration of aspartate was based on the previously published works in which up to 20 mM of aspartate was required to restore the proliferation of cells with impaired mitochondrial activity (Melendez-Rodriguez et al, 2019). Similarly, supplying EPA-treated PCa cells with 20 mM aspartate can stimulate cell proliferation (Fig 4). However, aspartate alone cannot bring the proliferation rate to the control group level, indicating that FO is a multi-target agent and can alter other proliferation-related pathways, as indicated by the GO enrichment and the KEGG results (Fig 2).

This study demonstrates the significant impact of dietary habits and nutrition on the growth of PCa cells and highlights the potential of $\omega$-3 PUFAs as therapeutic agents. By profiling a succinylome of PCa cells after FO treatment, the study showed evidence that $\omega$-3 PUFAs influenced the succinylation of a series of proteins associated with PCa cell proliferation, including the GOT2 protein. GOT2 succinylation may inhibit PCa progression through its interference with aspartate synthesis and nucleotide production. The results of this study provide a foundation for further research on succinylation and GOT2 as a potential drug target for future PCa treatments. Overall, this study sheds light on a potential treatment option for PCa disease, utilizing dietary and nutritional interventions.

# Materials and Methods

### Cell line and cell culture

Human androgen-independent PCa cell line PC-3 and human androgen-dependent PCa cell line LNCaP were purchased from the Shanghai Institute of Biochemistry and Cell Biology, Chinese

## A)

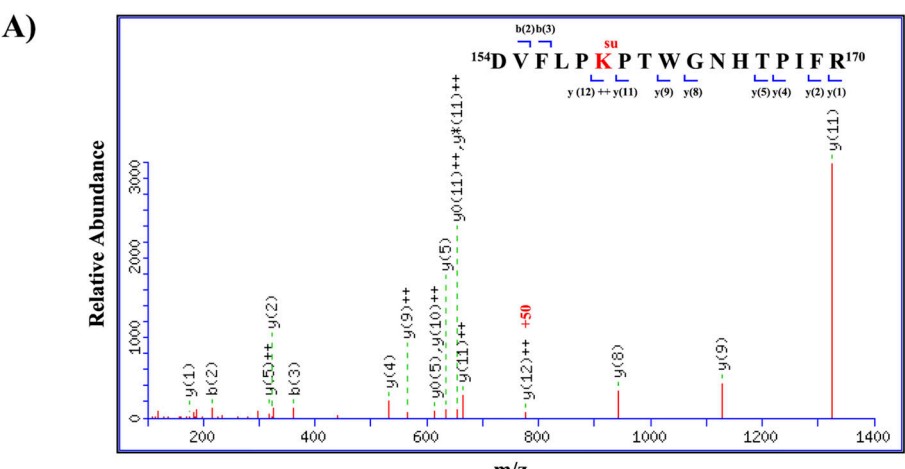

## B)

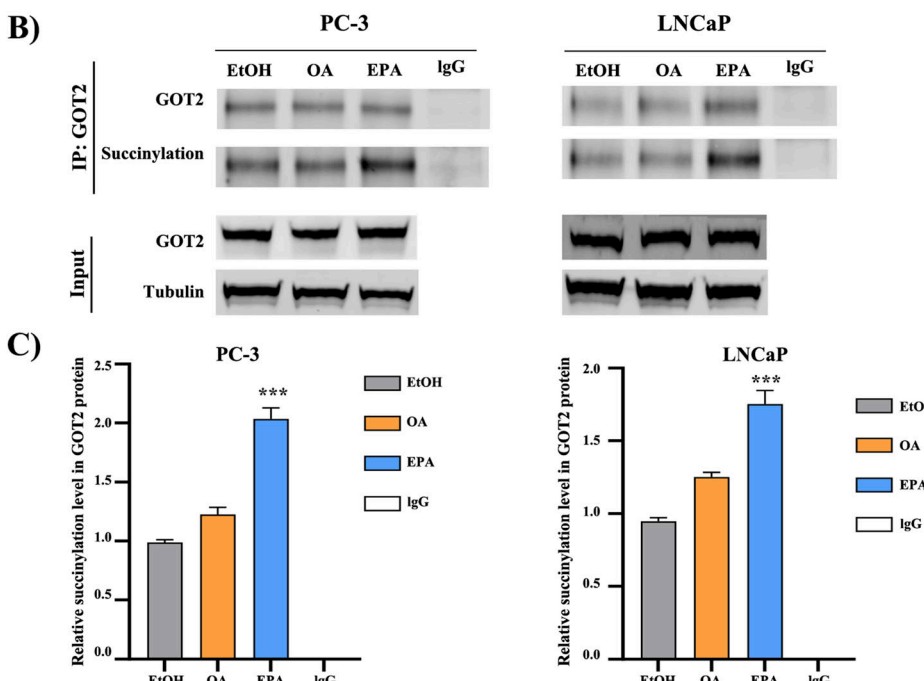

**Figure 3. Validation of GOT2 protein succinylation.**
**(A)** MS/MS spectrum of the detected succinylated peptide from the GOT2 protein. The specific succinylated peptide sequence is given; the red character indicates the succinylated lysine residue (K-159). **(B, C)** Immunoprecipitation (IP) results display the up-regulation of succinylation level after 100 $\mu$M EPA treatment. GOT2 antibody for IP is a mouse monoclonal antibody. Beyotime mouse lgG (A7028) as the negative control.
Source data are available for this figure.

Academy of Sciences. PC-3 cells are grown in F12 medium (cat# BL311A; Biosharp) with 10% FBS (Cat# 04-002-1ACS; Biological Industries), 1% penicillin and streptomycin (Cat# SV30010; Lab Genome). LNCaP is cultured in RPMI 1640 medium (Cat# BL303A; Biosharp) with 10% FBS (Cat# 04-002-1ACS; Biological industries) and 1% penicillin and streptomycin (Cat# SV30010; Lab Genome), supplied with 1% pyruvate (Cat# S8636-100ML; Sigma-Aldrich) and 1% L-glutamine (Cat# 25030081; Thermo Fisher Scientific). Both cell lines were incubated at 37°C in 5% $CO_2$.

### Cell proliferation experiments

#### Clonogenic assays
PCa cells were cultured in six-well plates (1,500 cells/well for PC-3, 4,000 cells/well for LNCaP, Cat# 3516; Corning). After 24 h, the same amount of EPA, OA or EtOH were added. PC-3 cells were continuously

cultured for 2 wk, and LNCaP cells for 3 wk before rapid Giemsa staining with solutions A and B (Cat# D011-3-3; Nanjing Jiancheng). The results were obtained after air drying the plates, and colonies were photoed with ChemiDoc MP (Bio-Rad). For data processing, Adobe Photoshop CC 2019 was used to count the number of colonies and Prism 8 software was used to summarize the results.

#### Growth curve
PCa cells were seeded into 24-well plates. After cells adhered to the plates (24 h for PC-3; 48 h for LNCaP), cells were treated with EPA, OA, and EtOH, respectively. Cell numbers were counted by Countess Automated Cell Counter (Cat# A49862; Thermo Fisher Scientific) in the following 6 d.

#### EdU cell proliferation experiments
PCa cells were cultured in 96-well plates pre-coated with poly-L-lysine at a density of 5,000 cells/well. After cells adhered to the

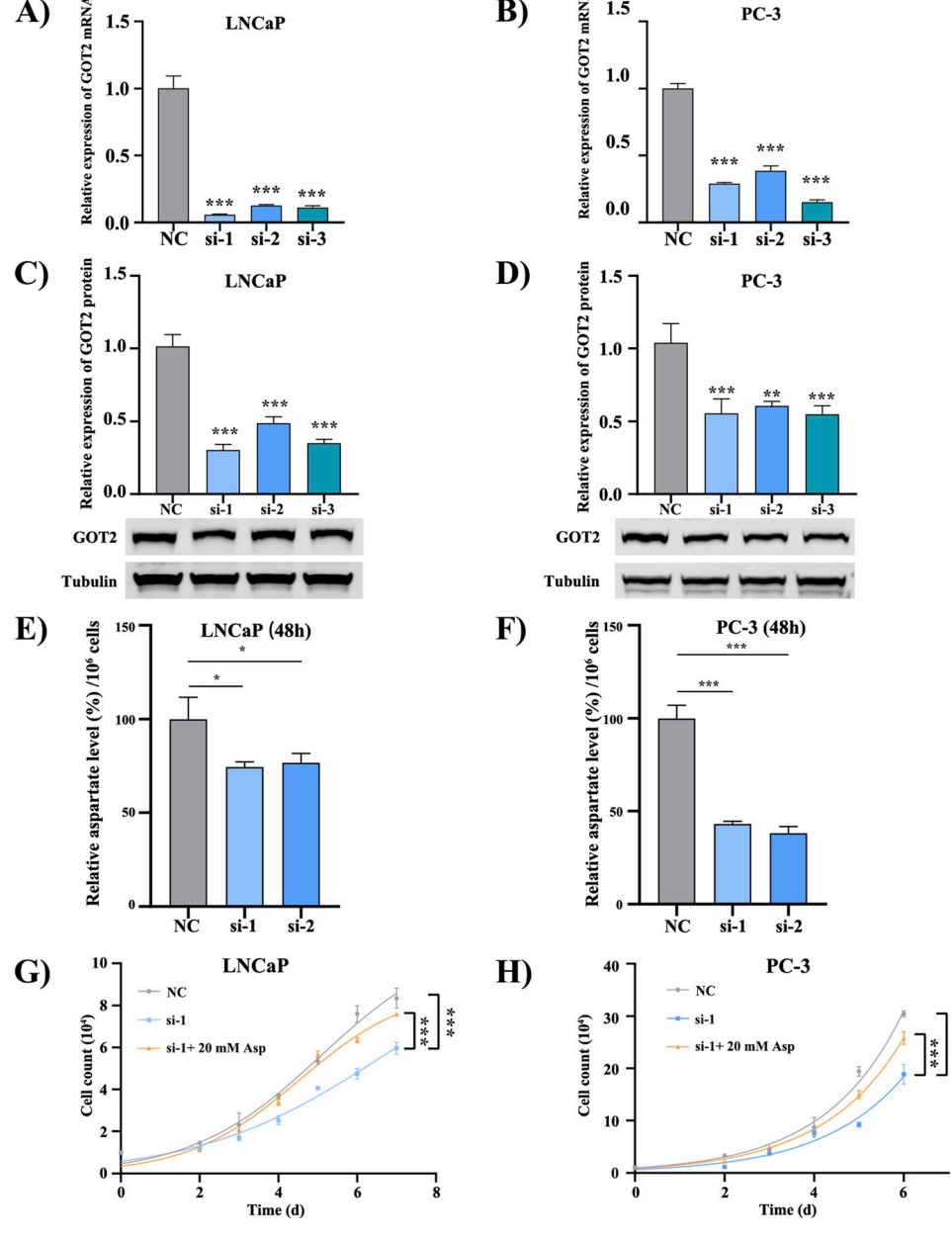

**Figure 4. GOT2 protein regulates cell proliferation by influencing cellular aspartate level.**
**(A, B)** qRT-PCR tests the knockdown efficiency in the RNA level after 24 h transfection. All three siRNA reduced the GOT2 mRNA level to less than 40%. **(C, D)** GOT2 protein level is down-regulated to less than half of its original level by designed siRNA after 48 h transfection. Tubulin is the indigenous control. **(E, F)** The aspartate assay confirmed the decline of aspartate contents after 48 h RNAi treatment in PCa cells (n = 3). All data were normalized by dividing the cell number. **(G, H)** Proliferation curves of PCa cells after GOT2 knockdown. si-1 and si-2: cells transfected by two different siRNAs targeting the GOT2 protein. NC is the negative control group; cells were transfected with a random siRNA sequence. Si-1+ 20 mM Asp and si-2+ 20 mM Asp: GOT2 down-regulated cells were supplied with 20 mM aspartate. PBS is the solvent control of Asp supplement. *P < 0.05; **P < 0.01; ***P < 0.001.
Source data are available for this figure.

bottom of the wells (24 h for PC-3; 48 h for LNCaP), treatments (EPA, OA, or EtOH) were applied. EdU (5-ethynyl-2′-deoxyuridine) assay was conducted using Beyoclick EdU Cell Proliferation Kit (Cat# C0088S; Beyotime) according to the manufacturer's instruction. The resulting absorbance was measured at 370 nm using SpectraMax i3x Multi-Mode Microplate Reader (Molecular Devices).

### Sample preparation for mass spectrometric analysis

#### *Protein extraction and trypsin digestion*
PCa cells were lysed first by 9 M urea (Cat# 29700; Thermo Fisher Scientific), followed by ultrasonic treatment (output power = 15 W; duration = 15 s). The supernatant was collected after centrifugation at

20,000*g*. BCA Assay Kit (Cat# P0009; Beyotime) was used to quantify the concentration of the collected protein samples. Urea lysate was used for concentration normalization. Then, 4.5 mM DTT (Cat# 7016; Cell Signaling Technology) and 10 mM iodoacetamide (Cat# I-6125; Sigma-Aldrich) were applied to reduce and alkylate samples. Tryptic digestion was carried out overnight with 10 μg/ml modified trypsin (Cat# V5113; Promega). The obtained peptides were acidified by 1% TFA (Cat# 28903; Thermo Fisher Scientific) and desalted by Sep-Pak Classic C18 Cartridge (SKU# WAT051910; Waters) before lyophilization.

#### *Enrichment of succinylated peptides*
After lyophilization, immunoaffinity purification (IP)-resuspended sample peptides were mixed with Succinyl-Lysine Immunoaffinity

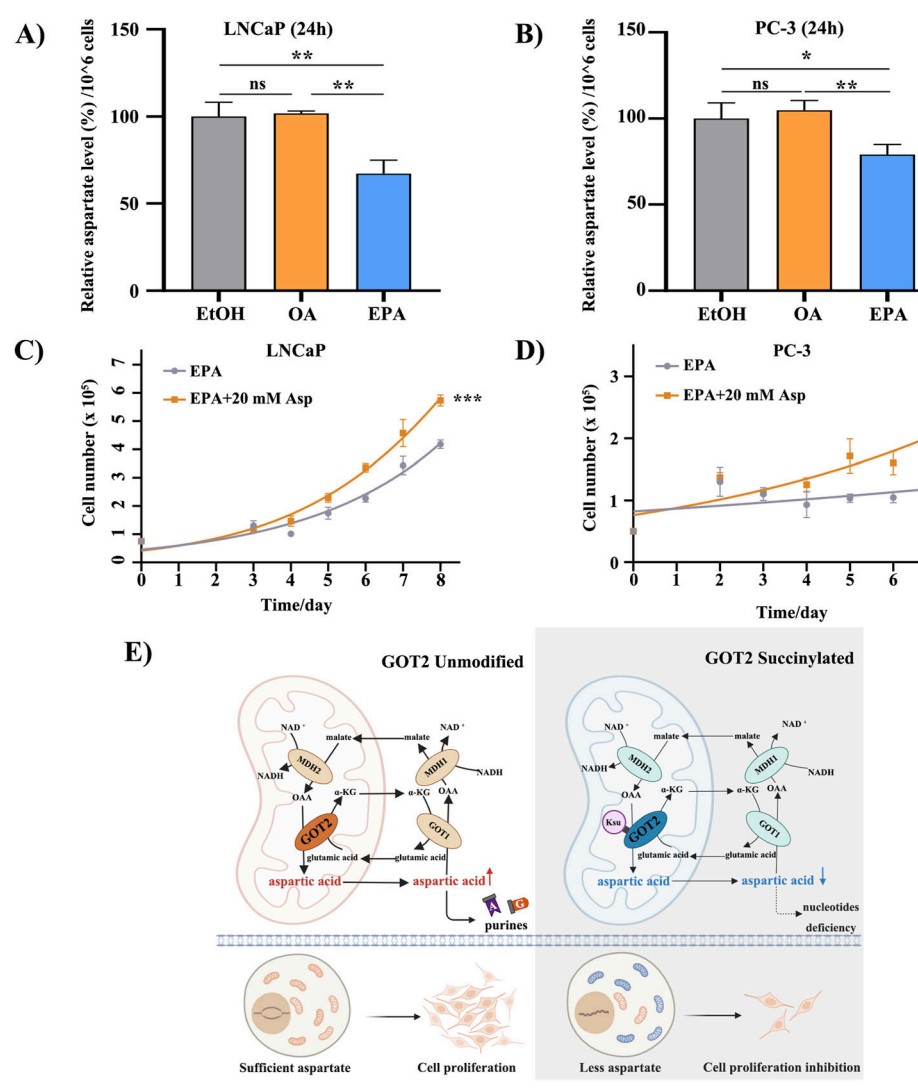

**Figure 5. FO inhibits PCa cell growth via succinylation of GOT2 and its interacting proteins.**
**(A, B)** The aspartate assay confirmed the decline of aspartate contents in PCa cells after 100 μM EPA treatment. The displayed data were normalized with cell number. **(C, D)** The proliferation curves of 100 μM EPA-treated PCa cells with or without 20 mM aspartate supplement in PCa cells. PBS is used as a solvent control. **(E)** Succinylation brings negative charges to its substrates. Hypothesized mechanism demonstrates identified succinylation involved in the malate–aspartate shuttle. Repulsive forces between GOT2 and its interacting proteins may lead to less aspartic acid and nucleotide generation (Created with BioRender.com, agreement number: TJ259OQ92W).

Beads (Cat# 13764; Cell Signaling Technology) and incubated for 2 h at 4°C. Enriched succinylated peptides eluted with 0.1% TFA (Cat# 28903; Thermo Fisher Scientific) and 40% acetonitrile (Cat# 51101; Thermo Fisher Scientific) were concentrated by SpeedVac Medium Capacity Vacuum Concentrators (Cat# SPD140DDA; Thermo Fisher Scientific).

## LC–MS/MS analysis

The NanoViper analytical column and pre-column were used for HPLC (Cat# DNV75150PN; Thermo Fisher Scientific). The Thermo Fisher Scientific EASY-nLC 1000 system separated the succinylated peptides, and the eluents were sprayed directly into the LTQ Orbitrap Elite mass spectrometer (Thermo Fisher Scientific) for analysis. The m/z range was set from m/z 300 to m/z 2,000, the eluting flow rate was 0.3 μl/min, the elution time was 180 min, and the capillary temperature was fixed at 275°C. A peptide with the sequence ALLGICQGGTGCYK was designed and synthesized by Apeptide for quality control purposes. Further details regarding instrument calibration and LC gradient conditions can be found in Tables S3 and S4.

## Data processing and analysis

### Raw data process
Raw data from LC–MS were preliminarily analyzed by MaxQuant_1.6.17.0. Because succinylation is not included in the default setting, it is added as a new PTM into the "Configuration" bar with the molecular formula $H(4) O(3) C(4)$. UP000005640 and its isoform file downloaded from the UniProt website (Accessed in 10/06/2022) were uploaded as reference sequences in the global parameter. Perseus software was utilized for data filtration, statistical analysis, and visualization. $P$ cut-off value smaller than 0.05 was considered significant.

### Bioinformatics analyses
Detected succinylated peptides were visualized using volcano plots generated with the ggplot2 R package. To further investigate their protein functions, human genome annotation data were

**Table 2.** siRNA sequences used in the GOT2 knock-down experiments.

| siRNA | Sequence (5′ to 3′) |
|---|---|
| si-1 | GCAUGCAGCUACAAGGUUATT |
| | UAACCUUGUAGCUGCAUGCTT |
| si-2 | UAACCUUGUAGCUGCAUGCTT |
| | GCAUGCAGCUACAAGGUUATT |
| si-3 | GCAACACAUCACCGACCAATT |
| | UUGGUCGGUGAUGUGUUGCTT |

obtained utilizing the org.Hs.eg.db R package. Subsequently, GO enrichment analysis and KEGG pathway analysis of peptides exhibiting significant changes in succinylation levels were conducted employing the clusterProfiler R package. The obtained results were ultimately refined and visually represented using BioRender (www.biorender.com). These analyses were implemented in R platform (version 4.1.1).

### Validation of MS results

#### Immunoprecipitation (IP)

PCa cells were treated for 24 h with EPA, OA, and EtOH before extracting protein. Because of the desuccinylase activity of some deacetylases, a deacetylase inhibitor cocktail, 100X (Cat# P1112; Beyotime), was mixed with the lysis buffer (Cat# P0013; Beyotime). The anti-AATM antibody (Cat# sc-271702; Santa Cruz Biotechnology) was used to enrich GOT2 protein from a 500-$\mu$g whole-protein mixture. Overnight, we used Protein A/G Sepharose beads (Cat# K286-25-5; Abcam) to capture the Fc fragment in the GOT2 antibody. After several PBST washes, the GOT2 protein was eluted by boiling with 2x SDS–PAGE loading buffer and collected by centrifugation (2,000$g$ for 2 min), followed by WB analysis using the anti-lysine succinylation motif antibody (Cat# PTM-419; PTM-BIO).

### Investigation of GOT2 protein function

#### GOT2 knockdown (RNAi)

LNCaP and PC-3 cells were transfected with siRNAs through Etta X-Porator H1 electroporator (Etta Biotech, Suzhou Industrial Park) to down-regulate GOT2 in both RNA and protein levels (voltage = 150 V; duration = 1500 $\mu$s; pulse number = 4; interval = 600 ms). Table 2 lists the siRNA sequence information.

**Quantitative real-time PCR (qRT-PCR)** Total RNA was extracted from cells using TRI Reagent (Cat# T3809-100ML; Sigma-Aldrich), and RT was performed with GoScript RT System (Cat# A5003; Promega) from 1 $\mu$g of extracted RNA. Target genes were amplified by qRT–PCR with BRYT Green Dye (Cat# A6001; Promega) binding. The cycle threshold (CT) values of targeted and housekeeping genes (tubulin) obtained by qRT–PCR were converted into signal intensities via the delta–delta method. GOT2 primer sequences are 5′-TTGCTGCTGCCATTCTGAAC-3′ (F) and 5′-GTCGGTGATGTGTTGCCAAT-3′ (R).

**Western blot analysis** The siRNA-transfected PCa cells were cultured for 48 h and protein lysates were collected using RIPA buffer

(Cat# R0278; Sigma-Aldrich) containing a protease inhibitor cocktail (Cat# P1005; Beyotime). Samples were quantified with BCA Assay Kit (Cat# P0012; Beyotime). 40 $\mu$g of protein per sample were loaded into the well of precast gel (Cat# M00657; GenScript). It was furthermore transferred to Immobilon-PSQ PVDF membranes (Cat# ISEQ00010; Merck) using the eBlot L1 Blotting System (Cat# L00686C; GenScript) and the corresponding kit (Cat# L00770C; GenScript). Transferred membranes were directly incubated with a 1:500 diluted primary GOT2 antibody (Cat# 14800-1-AP; Proteintech) overnight. The membranes were washed with TBS-Tween (TBST) followed by one hour of incubation with 680RD and 800CW infrared secondary antibodies (Cat# C50113-03 and C50512-05; LI-COR Biosciences). The membranes were washed in fresh TBST for 15–30 min then visualized using Odyssey Infrared Laser scanning imaging system (LI-COR).

**Aspartate quantification assay** The assay was conducted according to the manufacturer's instructions for the Aspartate Assay Kit (Cat# ab102512; Abcam). PCa cells were collected by trypsinization and ultrasonic digestion (output power = 15 W, duration = 5 s, interval = 5 s, pulse number = 12) in the aspartate assay buffer provided by the kit to release the aspartate content within the cells. Low-speed centrifugation (2,500$g$, 10 min) removed cell debris sediments. Results were obtained by measuring in OD 570 nm.

### Quantification and statistical analysis

The statistical analysis was carried out using GraphPad Prism 8. One-way ANOVA with Tukey's multiple comparison test and two-way ANOVA with Sidak's multiple comparison test were carried out when appropriate. Statistical significance is denoted in the figures as follows: *$P < 0.05$; **$P < 0.01$, ***$P < 0.001$.

## Data Availability

The datasets used and/or analyzed during the current study are available from the corresponding author on reasonable request.

## Supplementary Information

## Acknowledgements

We gratefully acknowledge the invaluable help from lab technicians Ms. Manting Qi and Ms. Shuxuan Zhao in sample collection and material purchasing. The authors would also like to thank Xi'an Jiaotong-Liverpool University (XJTLU) for FYP consumable funding support. Mass spectrometry resource was provided by XJTLU Centre for Pharmaceutical Analysis. This work was supported in part by XJTLU Key Programme Special Fund–KSF Exploratory Research KSF-E-14 (M Wang).

## Author Contributions

Y Jiang: conceptualization, data curation, formal analysis, validation, investigation, methodology, and writing—original draft, review, and editing.

C He: conceptualization, investigation, methodology, and writing—review and editing.

H Ye: resources, software, and validation.

Q Xu: data curation and formal analysis.

X Chen: conceptualization, data curation, and formal analysis.

Y Chen: data curation, formal analysis, supervision, validation, and methodology.

J Sun: supervision, validation, and methodology.

S Rocha: conceptualization, supervision, and writing—review and editing.

M Wang: conceptualization, resources, data curation, supervision, funding acquisition, validation, investigation, methodology, project administration, and writing—review and editing.

## Conflict of Interest Statement

The authors declare that they have no conflict of interest.

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
