## [Reviewer comments · Life Science Alliance]

Life Science Alliance

Comprehensive Analysis of the Lysine Succinylome in Fish Oil Treated Prostate Cancer Cells

Yifan Jiang, Chao He, Haokai Ye, Qianhan Xu, Xiuyuan Chen, Ying Chen, Jianying Sun, Sonia Rocha, and Mu Wang
DOI: <https://doi.org/10.26508/lsa.202302131>

Corresponding author(s): Mu Wang, Xi'an Jiaotong-Liverpool University

Review Timeline:

Submission Date:	2023-05-04
Editorial Decision:	2023-06-29
Revision Received:	2023-07-21
Editorial Decision:	2023-08-22
Revision Received:	2023-08-25
Accepted:	2023-08-28

Transaction Report:

June 29, 2023

Re: Life Science Alliance manuscript #LSA-2023-02131-T

Mu Wang
Department of Biological Sciences, Xi'an Jiaotong-Liverpool University

Dear Dr. Wang,

Thank you for submitting your manuscript entitled "Comprehensive Analysis of the Lysine Succinylome in Fish Oil Treated Prostate Cancer Cells" to Life Science Alliance. The manuscript was assessed by expert reviewers, whose comments are appended to this letter. We invite you to submit a revised manuscript addressing the Reviewer comments.

Thank you for this interesting contribution to Life Science Alliance. We are looking forward to receiving your revised manuscript.

Sincerely,

B. MANUSCRIPT ORGANIZATION AND FORMATTING:

Reviewer #1 (Comments to the Authors (Required)):

1. A short summary of the paper, including description of the advance offered to the field.

Brief Summary of the manuscript

In this study, the researchers focused on investigating the effects of fish oil (FO) and oleic acid (OA) on prostate cancer (PCa) cells. They found that FO, which is rich in omega-3 polyunsaturated fatty acids (ω -3 PUFAs), has a potential therapeutic value in PCa by disturbing mitochondria function in PCa cells.

Mitochondria play a crucial role in supporting cancer progression, for instance from the literature, studies have shown that the primary function of the electron transport chain (ETC) in cancer cells is to provide aspartate rather than ATP, and the malate-aspartate shuttle, mediated by glutamic-oxaloacetic transaminase 2 (GOT2), is involved in this process. Alterations to GOT2 can affect the generation of aspartate and nucleotides, which are essential for cellular proliferation.

The researchers aimed to understand the changes in mitochondrial physiological and chemical properties following FO treatment by focusing on Succinylation. This is a posttranslational modification where a succinyl group is added to a lysine residue of a protein molecule. Succinylation can modulate protein function by switching between acetylation and succinylation at specific lysine residues.

The researchers conducted a global profiling of succinylomic in PCa cells treated with FO, OA, and ethanol (EtOH) to identify succinylated peptides using mass spectrometry. Also, they investigated the differential protein succinylation levels in prostate cancer (PCa) cells under different physiological conditions. They categorized the succinylated proteins based on their function or location. To confirm the ability of eicosapentaenoic acid (EPA), a component of FO, to induce succinylation of GOT2, immunoprecipitation was performed on PCa cell lysate to enrich for GOT2 protein and assess its succinylation status. Aspartate quantification assays were conducted to validate the findings, and they demonstrated that normal GOT2 is responsible for generating increased levels of aspartate required for cellular proliferation.

The volcano plot of FO-EtOH and FO-OA revealed an overall upregulation of succinylation levels after FO treatment, suggesting the impact of FO on succinylation. The gene ontology (GO) analysis showed that succinylation primarily occurs in mitochondrial proteins, which can be attributed to the presence of succinyl-CoA, a donor of succinyl groups in the Krebs cycle. While GO-enriched pathways indicated a correlation between FO-induced succinylation of the GOT2 protein (glutamate oxaloacetate transaminase 2) and the malate-aspartate shuttle, which is involved in precursor metabolite generation. This finding suggested that FO treatment affects the production of aspartic acid, a crucial nitrogen donor for amino acid arginine, which is required for PCa progression.

The study theorized that FO-dependent structural changes and charge flips on the GOT2 K159 may lead to repelling forces, affecting aspartate production, and suppressing prostate cancer cell proliferation. Mitochondria play a vital role in nucleotide precursor production, and aspartate is necessary for robust cell proliferation. Previous research identified acetylation at the same site (K159), which enhanced protein-protein interactions within the malate-aspartate shuttle and promoted tumor growth. In this study, when GOT2 knockdown cells were provided with 20 mM aspartate in the media, it compensated for the aspartate deficiency caused by GOT2 knockdown. Similarly, supplying FO-treated PCa cells with 20 mM aspartate stimulated cell proliferation, although it did not fully restore the proliferation rate to the control group level. This suggests that FO affects multiple pathways related to cell proliferation.

In conclusion, this study suggests that FO can inhibit prostate cancer progression through multiple pathways and mechanisms. FO treatment influenced succinylation levels, particularly in mitochondrial proteins, and affected the malate-aspartate shuttle and aspartate production, thereby impacting PCa cell proliferation. However, FO's effects were not solely limited to aspartate metabolism, as indicated by the GO enrichment and KEGG pathway analysis. The findings highlight FO as a multi-target agent that can potentially alter various proliferation-related pathways.

This study offers a significant advance in the field by elucidating the molecular mechanisms through which dietary interventions, specifically ω -3 PUFAs, influence succinylation and impact PCa cell proliferation. The identification of GOT2 succinylation as a potential target for future PCa treatments provides valuable insights for the development of innovative therapeutic strategies. Thus, emphasizing the potential of dietary and nutritional approaches in the management and prevention of prostate cancer.

2. For each main point of the paper, please indicate if the data are strongly supportive. If not, explicitly state the additional experiments essential to support the claims made and the timeframe that these would require.

- i. Introduction- data are strongly supportive.
- ii. Methods and materials- data are strongly supportive.
- iii. Results and Discussion- data are strongly supportive.

iv. Conclusion - data are strongly supportive.

3. indicate any additional issues you feel should be addressed (text changes, data presentation, statistics etc.)

- Figure 1(C and D)- for x- axis labeling, write "days" in full. i.e., Time (days)

- Figure 3 (A) label the axes (y and x) of the spectrum.

Reviewer #2 (Comments to the Authors (Required)):

General Comment:

It is a pleasure to read the manuscript. The comprehensive experiments have demonstrated a mechanism at various levels. The authors had a good experimental design to tell a good story, but there are still many unknown questions. Can EPA (or FO) effectively treat PCa in clinic? As the authors already mentioned that this is only one of the mechanisms. How significant is this one for PCa treatment comparing to others on the impact on PCa? Could FO impact other cancer progression? Do other cancer cells have the same mechanism? Having said that, this research is good enough to be published with future clarifications mentioned below.

Specific Comment:

1. In the experiment of ω -3 PUFA EPA inhibits PCa cell growth, there was a nice dose response of EPA in colony number. What is the dose response of EPA in cell count? It seems no dose response of EPA in the EdU assay. What are the reasons? The performance of OA and EPA in colony number and cell counting is not consistent between LNCaP and PC-3 cells. How to explain it? Is OA supposed to increase the colony number, cell count, and relative EdU incorporation? What is the correlation among the three assays?

2. In the experiment of global succinylomic analysis of human PCa cell lines treated with FO, which dose groups of EPA were used? Among the 946 peptides, how many of them have a dose response? What standards did the authors used to obtain the 125 peptide changes? From EPA vs. OA, EPA vs. EtOH, and OA vs. EtOH? It is not clear from the text and figures.

3. The biosynthesis of amino acids involved 14 Ksu sites in 9 proteins. Why did the authors choose GOT2 rather than other? HSPD1 has the most Ksu sites, but it was not focused on the validation experiment. Could the authors explain the reasons?

4. What is the dose of EPA in Figure 3?

5. In Figure 4 and text, si-3 was mentioned nowhere except in Table 2. The authors should describe the si-3 before showing it in the figure. In Figures 4E-F, why is there no si-3 data? In Figures 4G-H, why is there no si-2 and si-3 data?

6. In Figure 5, what is the dose of EPA?

7. Since aspartate increases cell proliferation, should aspartate be a better drug target than GOT2 is?

Reviewer #3 (Comments to the Authors (Required)):

The manuscript by Jiang et al. reports lysine succinylation in PCa cells upon treatment with fish oil. Succinylation was first reported in 2011 but compared to phosphorylation, acetylation, methylation, and glycosylation, little is known about protein succinylation. In this context, authors attempt to map this PTM in PCa cells is important.

However, I have the following comments/concerns that needs to be addressed:

1) The representative succinylated spectrum shown in figure 3A is from Mascot but MaxQuant was used for data analysis.

2) I am not convinced that the Western blots from input and IP-GOT2 show differences in signal intensity among EtOH, OA and EPA in PCa cells (Figs. 3B, 4C, 4D).

3) The si-RNA KD seems not to have any depletion in GOT2 protein expression, though authors claim reduction in the protein level.

4) There is no description about the changes in succinylation sites as a function of treatments. How data were filtered. Were any filters were applied based on-site specific probability as done for phosphopeptides (e.g., class one succinylated residues with site specific probability score >0.75). This is important as there could be many false positives in this PTM analysis.

5) Details of LC-MS data acquisition are needed including instrument calibration, LC conditions, QCs, etc. Were succinylated peptides with 0 MS/MS counts removed?

Abstract:

In this study, PCa cell samples were injected into a mass spectrometry-based succinylomics platform to profile all detectable succinylated peptides.

Comments: Revise this as: This study employed a mass spectrometry-based approach to investigate succinylation in PCa cells. It highlights the potential of FO as a nutrition supplement for cancer therapy to slow down PCa progression.

Comments: Revise this as: This study highlights the potential use of FO as a nutrition supplement for managing and allowing down PCa progression.

Dear Editor and Reviewers:

Thank you for your correspondence and the reviewers' feedback regarding our manuscript titled "A Comprehensive Analysis of Lysine Succinylation in Prostate Cancer Cells Treated with Fish Oil."

The aforementioned comments are all of great value and provide significant assistance in revising and enhancing our manuscript. We have carefully reviewed all the comments and made revisions that we hope will be satisfactory. The revised portions are highlighted in yellow within the revised manuscript. The main corrections to the paper and our point-to-point responses to the reviewer's comments are outlined below:

Respond to the reviewer's comments:

Reviewer #1:

Thanks for your nice comments about our work, we have addressed your comment #3 and edited our Figures 1 and 3-5 accordingly to ensure the consistency in the axis.

Reviewer #2:

1. Can EPA (or FO) effectively treat PCa in clinic?

Response: FO is not currently an FDA-approved drug, but it is widely utilized as a nutritional supplement to prevent and inhibit or slow down cancer progression. In fact, preclinical evidence suggests a potential protective effect against the incidence and progression of prostate cancer; however, conflicting results have been reported in human studies (Aucoin et al. Integrative cancer therapies 2017). It is our hope that, through this study, we can identify some potentially potent new drug target(s) for prostate cancer therapy.

2. As the authors already mentioned that this is only one of the mechanisms. How significant is this one for PCa treatment comparing to others on the impact on PCa?

Response: The canonical approach to treating prostate cancer involves androgen deprivation therapy (ADT) in combination with docetaxel or abiraterone acetate, which is an aggressive method that only targets androgen-dependent PCa cells (in this study, we used LNCaP as a representative of this type of PCa cells) (Hahn *et al.* American Society of Clinical Oncology, 2018). For localized androgen-independent prostate cancer cells such as PC-3, treatment is more complex, typically involving

external beam radiotherapy (EBRT) or combination therapy (Chang *et al.* Nature reviews, 2014). In order to enhance treatment efficacy, FO is usually utilized as a dietary supplement in conjunction with traditional therapy. While EPA treatment may not be as effective as traditional methods for treating PCa, investigating how EPA inhibits PCa proliferation can provide valuable information about potential novel drug targets for PCa treatments - the ultimate goal of our research.

3. Could FO impact other cancer progression? Do other cancer cells have the same mechanism?

Response: Yes, with the exception of prostate cancer (PCa), FO has the ability to impede the progression of various types of cancer, including breast, colorectal (small and large intestine), lung, and colon cancer (Fabian *et al.* Breast cancer research, 2015; White *et al.* European journal of cancer prevention, 2019; Yang *et al.* Molecular carcinogenesis, 2014). The mechanism behind these effects is currently not completely understood; however, this research suggests that this may be associated with EPA's anti-inflammatory properties and lipid composition alternation. Previous studies have established a correlation between mitochondria and the inhibitory effect of FO on prostate cancer progression, however, they did not specifically investigate succinylation involvement. Therefore, our research's originality will provide additional insights.

4. In the experiment of ω -3 PUFA EPA inhibits PCa cell growth, there was a nice dose response of EPA in colony number. What is the dose response of EPA in cell count? It seems no dose response of EPA in the EdU assay. What are the reasons? The performance of OA and EPA in colony number and cell counting is not consistent between LNCaP and PC-3 cells. How to explain it? Is OA supposed to increase the colony number, cell count, and relative EdU incorporation? What is the correlation among the three assays?

Response: Thank you for your feedback. The colony assay serves as a reliable indicator of cell proliferation, as we closely monitored the cell count within the colonies during the experiment, considering only those with more than 50 cells. We appreciate your suggestion that the EPA dose-dependent growth curve assay could provide more precise cell number calculations. However, we find it unnecessary as the colony assay clearly demonstrates the inhibitory effect of EPA on PCa cell growth. Additionally, including too many trends in a growth curve may hinder its readability. Our goal is to ensure that our audience can easily discern how differently the same concentration of OA and EPA affect PCa cell growth.

The EdU assay results demonstrate the relative rate of proliferation indicated by thymidine analog incorporation. Although this result may not

be exactly dose-dependent, the trend is acceptable, with EPA inhibiting PCa cell proliferation. The coincidental observation in the LNCaP group is possible due to the limited treatment duration of 24 hours, which may not provide sufficient time to discern any significant trends.

As for the observed discrepancy between colony number and cell number, we attribute it to temporal factors. The colony formation assay requires a two-week duration, whereas the cell proliferation curve assay typically takes around one week. Additionally, the number of cells seeded for colony formation assay is significantly lower than that for cell growth assay. This may lead to excessive OA treatment and subsequent cell death. Conversely, the number of cells inoculated in the growth curve is more appropriate, allowing OA to promote cellular proliferation.

These three assays monitor distinct facets of cell proliferation. The colony formation assay provides an indication of proliferation rate by demonstrating the clonogenic potential of individual cells, whereas the growth curve assay necessitates extensive cell counts and highly skilled training to minimize errors and large standard deviations from the counts (Morten *et al.* Journal of visualized experiments, 2016). However, these aforementioned methods can only assess apparent cell growth and cannot indicate the speed of DNA replication. Therefore, we utilized the EdU assay, an acknowledged gold standard, to address this limitation (Angelozzi *et al.* Methods in Molecular Biology, 2021). These interesting findings then leads to our following research.

- 5. In the experiment of global succinylomic analysis of human PCa cell lines treated with FO, which dose groups of EPA were used? Among the 946 peptides, how many of them have a dose response? What standards did the authors used to obtain the 125 peptide changes? From EPA vs. OA, EPA vs. EtOH, and OA vs. EtOH? It is not clear from the text and figures.**

Response: The MS analysis used a concentration of 100 μ M EPA, which has been added to page 6 with changes highlighted in red font. Thank you for reminding us about this missing information

In this study, we refrained from conducting the dose-dependent experiment due to several reasons. In mass spectrometry, response or signal intensity is commonly utilized for relative quantification; however, this method may not be entirely accurate, as any subtle change in the environment will influence the outcome of MS. The ionization efficiency can also affect the mass spectrometry response intensity, and it becomes challenging to detect ions that undergo neutral loss.

We selected 125 peptides from a pool of 946 peptides by excluding the unsuccinylated ones.

6. The biosynthesis of amino acids involved 14 Ksu sites in 9 proteins. Why did the authors choose GOT2 rather than other? HSPD1 has the most Ksu sites, but it was not focused on the validation experiment. Could the authors explain the reasons?

Response: As our previous research has demonstrated the inhibitory effect of EPA on PCa cells, it is noteworthy that the significance of HSPD1 in cell proliferation is not as pronounced as that of GOT2. Additionally, we have specifically chosen the GOT2 protein over other amino acid biosynthesis-related proteins due to its unique contribution in nucleotide biosynthesis. Furthermore, prior studies have investigated the acetylation of K159 site within the GOT2 protein and its role in regulating protein-protein interactions that facilitate aspartate production and cancer progression (Xiao *et al.* Stroke 2021). Therefore, we posit that GOT2 plays a pivotal role in the progression of prostate cancer, and our ongoing research is focused on elucidating the precise mechanisms underlying its involvement in mitochondrial function.

7. What is the dose of EPA in Figure 3? In Figure 5, what is the dose of EPA?

Response: The EPA concentration employed in all figures, except for figure 1, was set at a concentration of 100 μ M. We have made certain revisions to the figure legend on pages 25-26 in order to improve its clarity and precision. We appreciate your bringing this matter to our attention.

8. In Figure 4 and text, si-3 was mentioned nowhere except in Table 2. The authors should describe the si-3 before showing it in the figure. In Figures 4E-F, why is there no si-3 data? In Figures 4G-H, why is there no si-2 and si-3 data?

Response: Thank you for your suggestions. The initial inclusion of three siRNAs because we synthesized three effective knockdowns targeting the GOT2 protein, and we selected two out of the three that consistently performed well. We have chosen to only present the si-1 data in our manuscript because the inclusion of si-2 data would result in a cluttered graph and therefore has been omitted from the figure. To provide more information, we have included the si-2 data in the additional file 1 as S6 Fig, and corresponding revisions have been made on page 8 of the manuscript.

A) LNCaP cell proliferation curve with GOT2 knockdown and L-aspartate supplement

B) PC-3 cell proliferation curve with GOT2 knockdown and L-aspartate supplement

****: the difference between si-1 and si-1 + 20mM Asp

#####: The difference between si-2 and si-2 + 20mM Asp

9. Since aspartate increases cell proliferation, should aspartate be a better drug target than GOT2 is?

Response: While SLC1A3-mediated importation of aspartate from the culture medium is significant (Garcia-Bermudez *et al.* Nat Cell Biol 2018; Sun *et al.* The EMBO journal 2019), it cannot fully meet the demand of cancer cells that heavily rely on mitochondrial production of aspartate. Therefore, targeting the mitochondrial GOT2 protein could be an innovative and potentially effective approach.

Reviewer #3:

1. **The representative succinylated spectrum shown in figure 3A is from Mascot but MaxQuant was used for data analysis.**

Response: The Mascot software outperforms MaxQuant in terms of spectrum presentation due to its superior readability. Furthermore, discrepancies in MS spectra do not compromise data quality or reliability, thus we opted for a software that enhances audience comprehension.

2. **I am not convinced that the Western blots from input and IP-GOT2 show differences in signal intensity among EtOH, OA and EPA in PCa cells (Figs. 3B, 4C, 4D).**

Response: The treatments of EtOH, OA and EPA did not induce any alterations in the protein levels of GOT2, but led to significant changes in succinylation. We hypothesize that it is not the GOT2 protein itself undergoing a change in levels, but rather the succinylation on it that leads to decrease in nucleotide biosynthesis.

3. **The si-RNA KD seems not to have any depletion in GOT2 protein expression, though authors claim reduction in the protein level.**

Response: Based on the grey value provided by Image Studio software (Version 5.2), we have calculated a significant decrease in its level, as indicated by the bar chart. To provide more information, we have included the WB original data in the additional file 1 as S5 Fig, and corresponding revisions have been made on page 8 of the manuscript.

- There is no description about the changes in succinylation sites as a function of treatments. How data were filtered. Were any filters were applied based on-site specific probability as done for **succinylpeptides** (e.g., class one succinylated residues with site specific probability score >0.75). This is important as there could be many false positives in this PTM analysis.

Response: We are currently introducing mutations to mimic succinylated sites in GOT2 in order to investigate the functional implications of succinylation at these specific locations. Our selection criteria for these residues were based on a site-specific probability score threshold of >0.75.

- Details of LC-MS data acquisition are needed including instrument calibration, LC (gradient) conditions, QCs, etc. Were succinylated peptides with 0 MS/MS counts removed?

Response:

We have incorporated the following information in our manuscript on page 14 and in Table S3-4 of Additional File 2.

Instrument calibration:

Software	LTQ Tune
calibration solution	LTQ VELOS ESI Positive Ion Calibration Solution (Thermo-Fisher Scientific, Cat# 88323)
Heater Temp	Room temperature
Sheath Gas Flow Rate	0-5
Aux Gas Flow Rate	0
Sweep Gas Flow Rate	0
I Spray Voltage	3-3.8
Capillary Temp	275
Tube Lens	30

The flow rate of calibration solution: 1-5 μ L/min

LC gradient condition:

Time	duration	Flow [nL/min]	%B (acetonitrile)
00:00	N/A	300	5
02:00	02:00	300	5
152:00	150:00	300	50
157:00	05:00	300	95
162:00	05:00	300	95
172:00	10:00	300	5
182:00	10:00	300	5

QCs: We designed a peptide as a standard (sequence: ALLGICQGGTGCK, Apeptide, Shanghai, China).

The succinylated peptides with 0 MS/MS counts were removed.

6. Abstract:

In this study, PCa cell samples were injected into a mass spectrometry-based succinylomics platform to profile all detectable succinylated peptides.

Comments: Revise this as: This study employed a mass spectrometry-based approach to investigate succinylation in PCa cells.

It highlights the potential of FO as a nutrition supplement for cancer therapy to slow down PCa progression.

Comments: Revise this as: This study highlights the potential use of FO as a nutrition supplement for managing and **slowing** down PCa progression.

Response: Thank you for your suggestion. We have revised our abstract on page 2 by incorporating your recommended language.

We tried our best to improve the manuscript and made some changes in the manuscript. These changes will not influence the content and framework of the paper. And here we did not list the changes but marked in red in revised paper. We appreciate for Editors/Reviewer's warm work earnestly, and hope that the correction will meet with approval.

Once again, thank you very much for your comments and suggestions.

August 22, 2023

RE: Life Science Alliance Manuscript #LSA-2023-02131-TR

Prof. Mu Wang
Xi'an Jiaotong-Liverpool University
Academy of Pharmacy
111 Ren'ai Road
PB236
Suzhou, Jiangsu 215123
China

Dear Dr. Wang,

Thank you for submitting your revised manuscript entitled "Comprehensive Analysis of the Lysine Succinylome in Fish Oil Treated Prostate Cancer Cells". We would be happy to publish your paper in Life Science Alliance pending final revisions necessary to meet our formatting guidelines.

- please make sure the author order in your manuscript and our system match
- please consult our manuscript preparation guidelines <https://www.life-science-alliance.org/manuscript-prep> and make sure your manuscript sections are in the correct order
- please upload your main and supplementary figures as single files; - Please upload all figure files as individual ones, including the supplementary figure files; all figure legends should only appear in the main manuscript file
- please add your main, supplementary figure, and table legends to the main manuscript text after the references section
- please add a callout for Fig 2A, Fig 2B, Fig 4E, Fig 4F, Fig 5C, Fig 5D, Fig S1A, Fig S1B, Fig S5A, Fig S5B to your main manuscript text;
- please upload your Tables in editable .doc or excel format
- please incorporate any points from the Conclusion section into the Discussion, we only allow a Discussion section

A. FINAL FILES:

B. MANUSCRIPT ORGANIZATION AND FORMATTING:

Sincerely,

Reviewer #2 (Comments to the Authors (Required)):

The authors have made significant clarifications to address the questions. The manuscript provides unique value for prostate cancer treatment. I recommend that it should be accepted for the publication.

Reviewer #3 (Comments to the Authors (Required)):

Authors have addressed all my previous comments satisfactorily, and recommend for its acceptable in the current form.

August 28, 2023

RE: Life Science Alliance Manuscript #LSA-2023-02131-TRR

Prof. Mu Wang
Xi'an Jiaotong-Liverpool University
Academy of Pharmacy
111 Ren'ai Road
PB236
Suzhou, Jiangsu 215123
China

Dear Dr. Wang,

Thank you for submitting your Research Article entitled "Comprehensive Analysis of the Lysine Succinylome in Fish Oil Treated Prostate Cancer Cells". It is a pleasure to let you know that your manuscript is now accepted for publication in Life Science Alliance. Congratulations on this interesting work.

DISTRIBUTION OF MATERIALS:

Again, congratulations on a very nice paper. I hope you found the review process to be constructive and are pleased with how the manuscript was handled editorially. We look forward to future exciting submissions from your lab.

Sincerely,
